# Saffron as a Promising Therapy for Inflammatory Bowel Disease

**DOI:** 10.3390/nu16142353

**Published:** 2024-07-20

**Authors:** Mudasir Rashid, Rumaisa Rashid, Sabtain Saroya, Mrinalini Deverapalli, Hassan Brim, Hassan Ashktorab

**Affiliations:** Department of Medicine and Cancer Center, Howard University College of Medicine, Washington, DC 20059, USA; mudasir.rashid@howard.edu (M.R.); rumaisa.rashid@howard.edu (R.R.); sabtain.saroya@bison.howard.edu (S.S.); mrinalini.deverapal@howard.edu (M.D.); hbrim@howard.edu (H.B.)

**Keywords:** IBD, CD, UC, calprotectin, saffron

## Abstract

Inflammatory bowel disease (IBD) is a chronic inflammatory illness of the gastrointestinal tract (GI), characterized by recurrent episodes of inflammation and tissue destruction. It affects an increasing number of individuals worldwide who suffer from Crohn’s disease (CD) or ulcerative colitis (UC). Despite substantial advances in understanding the underlying causes of IBD, the available treatments remain restricted and are sometimes accompanied by severe consequences. Consequently, there is an urgent need to study alternate therapeutic options. This review assesses the present drugs, identifies their limitations, and proposes the use of saffron, a natural plant with great therapeutic potential based on preclinical and clinical investigations. Saffron has gained attention for its potential therapeutic benefits in treating various ailments due to its established bioactive compounds possessing antioxidant and anti-inflammatory properties. This review covers how saffron impacts the levels of calprotectin, an inflammatory marker, for various inflammatory responses in multiple diseases including IBD. Data from clinical trials were assessed to determine the efficacy and safety of using saffron to counter inflammation in multiple diseases. Studies have shown that saffron may protect against inflammatory bowel disease (IBD) through several mechanisms by inhibiting pro-inflammatory cytokines (TNF-α, IL-1β, and IL-6), reducing oxidative stress through antioxidant effects, enhancing mucosal barrier function by upregulating tight junction proteins, and modulating the gut microbiota composition to promote beneficial bacteria while suppressing pathogenic ones; these combined actions contribute to its therapeutic potential in managing and alleviating the symptoms of IBD. This will enable future research endeavors and expedite the translation of saffron-based interventions into clinical practice as a valuable adjunctive therapy or a potential alternative to conventional treatments, thereby enhancing the quality of life for individuals suffering from inflammatory diseases including IBD.

## 1. Introduction

Inflammatory bowel disease (IBD) is a chronic inflammatory illness of the gastrointestinal (GI) tract that affects millions of people worldwide as either Crohn’s disease (CD) or ulcerative colitis (UC). The UC diagnosis was first established in 1859 [1], whereas CD was first defined in 1932 [2]. The pathogenesis of IBD involves a complicated interplay between genetic, environmental, and immunological variables, including the gut microbiome and opportunistic pathogens [3,4]. IBD is a complicated and multifaceted disorder, associated with different symptoms, such as abdominal pain, diarrhea, exhaustion, and weight loss [5,6].

CD and UC exhibit a spectrum of clinical and pathological features. The inflammatory location and nature distinguish the two phenotypes (shown in Figure 1). While UC affects only the colon mucosa, CD can impact any section of the GI tract [7]. CD and UC have similar clinical aspects, including extra-intestinal symptoms. However, hematochezia and passing mucus or pus are more frequent in UC, while CD frequently causes fistulas, perianal disease, and colonic and small intestinal blockages. Cryptitis and crypt abscesses are common in both disorders; however, the crypt architecture is more deformed in UC [8]. Both UC and CD demonstrate recurrent intestinal inflammation. Intermediate colitis (IC) may not necessarily present different clinical symptoms to either UC or CD, which can make it difficult to discriminate between the two. While UC is not a discrete illness or a clinical entity, it accounts for roughly 10% of all colon-related IBD cases [9], and this has not changed over the last 30 years [10]. CD is characterized by sporadic transmural inflammation that can arise in any section of the digestive tract, while UC demonstrates chronic and more surface-level inflammation of the mucosa and submucosa, typically localized to the colon. In CD, inflammation typically affects the lower part of the small intestine (referred to as ileal CD, accounting for around 80% of cases) and the colon (known as Crohn’s colitis, seen in about 30% of cases). Conversely, inflammatory lesions in UC develop in the lower portion of the colon (UC proctitis), although they can extend throughout the entire colon (UC pancolitis, observed in 20% of cases). The clinical features commonly associated with CD and UC are listed in Figure 1.

Although the exact cause and etiology of IBD are still unknown, it is believed to occur due to an abnormal immune response and environmental factors that result in long-lasting chronic inflammation in the digestive tract [11,12].

Furthermore, novel therapies for IBD are expanding, offering new hope but also facing limitations such as variable patient response and high costs. Calprotectin, a biomarker of inflammation, is crucial for monitoring disease activity and guiding treatment decisions. Several lines of evidence have been emerging, highlighting saffron’s potential as an IBD therapy due to its anti-inflammatory and antioxidant properties, and it has been found to reduce inflammation and oxidative stress, key factors in IBD pathogenesis. Early studies indicate that saffron offers a natural, cost-effective alternative or complement to existing IBD treatments, promising a high efficacy and safety based on clinical validation, and has been shown to alleviate symptoms and improve intestinal health. The aim of this review article was to collect information about IBD pathogenesis, type, conventional treatment, and saffron (its active compounds) in terms of preclinical and clinical observations. Additionally, we highlighted that saffron is safe and cost-effective with minimal potential side effects, and, finally, we discussed our perspective on saffron use as an adjuvant therapy for IBD that improves patient outcomes and personalizes IBD management.

## 2. Current Treatments and Their Limitations

There is currently no cure available for IBD, and treatment primarily involves managing symptoms and preventing further complications. The treatment outcomes also depend on the type and severity of the disease [13,14]. Some of the currently available IBD treatments include amino-salicylates, corticosteroids, immunomodulators, biologics, surgery, and lifestyle changes [15,16,17]. The following pharmacological and non-pharmacological interventions were used for IBD:

I.Pharmacologic interventions:

**A. Amino-salicylates** (ASAs, also known as 5-Amino-Salicylic Acid or 5-ASA) are a class of drugs commonly used to reduce inflammation in the digestive tract and can be administered orally or rectally, depending on the location and severity of the inflammation [18,19]. The most common FDA-approved ASAs used in the treatment of IBD are sulfasalazine, mesalamine, and balsalazide [15]. In 2004, the British Society of Gastroenterology guidelines for UC management considered 5-ASA’s mechanism of action, delivery, and maintenance strategies [20,21,22,23]. Several mechanisms of action have been reported that act locally to inhibit the activity of prostaglandins and leukotrienes (one of the major factors involved in inflammatory response) in the gut. Other mechanisms are as follows: (I) Anti-inflammatory effects: It has been shown that ASAs reduce inflammation in the GI tract by inhibiting the production of pro-inflammatory cytokines such as tumor necrosis factor-alpha (TNF-α) and interleukin-1 (IL-1). (II) Antioxidant effects: ASAs have been shown to have antioxidant properties, which help to protect cells in the GI tract from oxidative stress. (III) Immune modulation: ASAs have been shown to modulate the immune response in the GI tract by reducing inflammation and preventing tissue damage [22,24,25,26]. However, the exact mechanism of action is not completely understood and needs to be deciphered. Apart from the beneficial effects, several ASA-associated side effects have been reported, such as nausea, vomiting, diarrhea, headache, abdominal pain, dizziness, and, in rare cases, more serious side effects, such as allergic reactions, liver damage, and kidney damage [21,27,28]. It is important to note that ASAs are considered to be a mild treatment for IBD and may not be effective for more severe or widespread disease. Reports have shown that ASAs, in some cases, may not be effective, or their effectiveness may diminish over time due to inadequate dosing or poor absorption, disease severity and location, individual variation in responses, and drug resistance. Therefore, combining therapies such as corticosteroids, immunosuppressants, and/or biologics with ASA might be needed to enhance the control of inflammation in certain patients.

**B. Corticosteroids** are a class of drugs that includes prednisone, budesonide, hydrocortisone, and dexamethasone, which can be administered orally or via injection depending on the severity and location of the inflammation. These are used for the short-term management of acute flares and are not recommended for long-term use due to their potential for significant side effects [15,29,30,31,32]. Several studies have shown that the mode of action of corticosteroids is the suppression of the immune system’s response through binding to specific receptors on immune cells and inhibiting the production of inflammatory molecules (cytokines and prostaglandins), thus reducing inflammation and helping to alleviate symptoms such as pain, diarrhea, and rectal bleeding in patients with IBD [33]. In addition, corticosteroids have been shown to aid in decreasing the severity of inflammation in IBD by reducing the number of white blood cells circulating in the blood, including neutrophils, lymphocytes, and eosinophils, which are involved in the inflammatory response [34,35].

Potential side effects include increased appetite; weight gain; mood changes; sleep disturbances; increased risk of infections; high blood pressure; high blood sugar; and, in rare cases, osteoporosis, glaucoma, and adrenal insufficiency [36,37]. Corticosteroids may not be effective for all patients, or they may need higher doses or additional therapies to achieve and maintain remission, followed by a transition to biologic therapy. Reports have shown that corticosteroids may not be effective in treating IBD, due to the disease’s severity, its duration, individual variation, and patients’ non-adherence to the medication regimen. Therefore, combination therapy with corticosteroids and other medications, such as ASAs, immunosuppressants, or biologics, may be needed for the optimal control of inflammation.

**C. Immunosuppressants** are a class of medications used in the long-term management of IBD that work by suppressing the immune system to reduce inflammation in the gut by targeting specific immune cells and pathways and inhibiting the production of DNA and RNA (the inhibition of purine synthesis and the suppression of immune cell proliferation), which slows down the proliferation of immune cells (inhibits the activity of T cells and involves the suppression of T-cell activation and cytokine production) [38,39,40]. Commonly used immunosuppressants include azathioprine (Imuran), mercaptopurine (Purinethol), methotrexate (Rheumatrex), cyclosporine (Sandimmune), and tacrolimus (Prograf) [41,42,43]. Reports have shown that immunosuppressants have side effects such as nausea, vomiting, diarrhea, liver toxicity, increased risk of infection, risk of cervical high-grade dysplasia, increased risk of lymphoproliferative disorders, and skin cancers [44,45,46]. Reports have shown that immunosuppressants help maintain remission and reduce the need for corticosteroids, but they take weeks to months to become effective, and regular monitoring for potential side effects in patients with IBD is needed [47,48]. Reports have shown that, in some cases, immunosuppressants may not be effective in treating IBD or may only provide a partial relief of symptoms due to the disease’s severity and duration, individual variation in responses, resistance to treatment, non-adherence to treatment, inflammation driven by non-immune mechanisms, genetic susceptibility, gut microbiota dysbiosis, and environmental triggers that also contribute to inflammation in IBD [28,49,50]. Therefore, combinatorial therapy and careful consideration are required when using immunosuppressants to treat patients with IBD.

**D. Biologics** are a class of medications given via intravenous or subcutaneous infusion that can be effectively used in reducing inflammation in IBD by targeting specific proteins involved in the immune response [51,52,53]. The choice of these agents depends on the nature and phenotype of IBD, prior biologic exposure and response, patient comorbidities, and the potential adverse effects of the therapy. The current FDA-approved biologic agents or drugs commonly used to treat CD and UC are (I) anti-TNF agents, such as infliximab (chimeric monoclonal antibody), adalimumab, and certolizumab pegol (a fully human monoclonal antibody), which bind to and neutralize or block the activity of TNF-α, a pro-inflammatory cytokine that plays a key role in the pathogenesis of IBD; (II) integrin inhibitors (vedolizumab and natalizumab), monoclonal antibodies that block the activity of α4β7 integrin (involved in the trafficking of lymphocytes to the gut); (III) IL-12/23 inhibitors, such as ustekinumab, which is a monoclonal antibody that binds to and blocks the activity of IL-12 and IL-23 (cytokines play key roles in the pathogenesis of IBD); (IV) IL-17 inhibitors, monoclonal antibodies (ixekizumab and secukinumab) that bind to and block the activity of IL-17, a cytokine involved in the immune system’s inflammatory response; and (V) JAK inhibitors (tofacitinib and upadacitinib), which block the activity of Janus kinases 1/3 (JAK1 and JAK3 enzymes), which play a role in the immune system’s inflammatory response, resulting in a reduction in inflammation in the GI tract. JAK inhibitors are typically used when other treatments for IBD, such as corticosteroids or immunosuppressants, have been unsuccessful. Anti-TNF therapy effectively maintains clinical remission and reduces the surgery risks in IBD beyond one year, with deep remission patients at a lower risk of relapse after withdrawal [54,55,56].

However, biologics have potential side effects such as infusion reactions, an increased risk of infections (including the reactivation of latent tuberculosis), sepsis, injection-site reactions, headache, fatigue, abdominal pain, nausea, and an increased risk of a rare but serious brain infection called progressive multifocal leukoencephalopathy (PML). In rare cases, there are other side effects, such as blood clots, various cancers (lung, breast, leukemia, skin, and prostate) [57,58,59], fever, chills, rash, an increased risk of developing other autoimmune disorders (lupus and multiple sclerosis), severe allergic reactions (anaphylaxis), and neurological symptoms (multiple sclerosis-like symptoms and peripheral neuropathy) [60,61,62]. Biologic agents can effectively prevent severe complications of IBD when used early; however, it is important to note that the risk and severity of the side effects can vary depending on the specific biologic medication, the patient’s individual characteristics, and the length of treatment. The drug’s effectiveness, safety, availability, and price, and patient preferences must be considered before initiating therapy [63,64]. Appendix A summarizes the IBD drug treatments and side effects.

II.Non-pharmacologic interventions:

**A. Surgery** is considered when medication and other treatments are not effectively managing the symptoms or if there are complications such as strictures, abscesses, or fistulas [65,66]. Several studies have shown that the choice of surgical procedures such as colectomy, proctocolectomy, ileostomy, and ileal pouch–anal anastomosis (IPAA) for each disease (CD and UC) varies [66,67,68]. For example, in patients with UC, colectomy may be performed, which involves the removal of the entire colon and is typically recommended if medication and other treatments have not been effective or if a substantial risk of colon cancer is suspected. Depending on the specific UC case, the surgeon may also remove the rectum and anus (proctocolectomy), which can be performed in either one or two stages, with a temporary ileostomy in between [65,66,69]. In patients with CD, surgery is usually performed to treat specific complications, such as strictures, abscesses, or fistulas, which may require the removal of a portion of the intestine, while, in other cases, there is a need to create an ostomy (creating an opening in the abdomen to allow waste to pass through a bag). The cumulative risk for colorectal cancer in patients with UC is 2% after 10 years, 8% after 20 years, and 18% after 30 years of diagnosis [70,71,72]. In some cases, surgery may involve removing the affected part of the intestine or colon and the rectum, which is often seen as a curative option depending on whether it is UC or CD [73,74]. The side effects of surgical procedures include infection, bleeding, bowel obstruction, and the need for lifelong management. Additionally, reports have shown that surgery does not address the underlying immune dysfunction; however, it improves the quality of life to a certain extent in some patients.

**B. Lifestyle changes:** IBD is one of the diseases associated with lifestyle risk factors such as diet, stress, lack of exercise, smoking, drugs such as NSAIDs, sleep, and alcohol consumption. Several reports have shown that several lifestyle changes may be recommended to reduce inflammation and improve overall health and well-being for individuals with IBD [13,14]. **(I)** Diet: Studies have reported that dietary changes—such as avoiding foods that trigger inflammation and exacerbate IBD symptoms; increasing fiber intake; and limiting sugar, refined carbohydrates, and saturated and trans fat intake—are beneficial. Other reports have shown benefits from a low-FODMAP diet, a gluten-free diet, or a specific carbohydrate diet (SCD) [75,76,77]. **(II)** Stress reduction: Reports have shown that stress does not cause IBD; however, it can worsen symptoms and trigger flare-ups in people who already have the condition. There is evidence showing that different relaxation techniques such as deep breathing, meditation, and yoga can help calm the mind and reduce stress levels, partly managing IBD symptoms [78,79,80,81]. **(III)** Exercise: Evidence suggests that regular exercise can improve the symptoms of IBD and reduce inflammation via increasing energy levels, reducing stress, and promoting a healthy immune system [82,83,84]. Furthermore, studies have also shown that regular exercise induces the release of anti-inflammatory cytokines, which can reduce inflammation in the body, including in the gut [82,85,86]. **(IV)** Smoking cessation: Several studies have demonstrated that smoking worsens IBD symptoms and increases the risk of complications, and quitting smoking has beneficial effects in reducing inflammation and improving overall health in patients with IBD [87,88,89]. Furthermore, several studies have shown that smoking can have a negative impact on the immune system by affecting the gut lining through increasing the permeability of the intestinal wall, allowing bacteria and other harmful substances to enter the bloodstream. This leads to an overactive immune response and can increase inflammation and worsen symptoms. **(V)** NSAID use: Studies have demonstrated that nonsteroidal anti-inflammatory drugs (NSAIDs) such as aspirin and ibuprofen can irritate the intestinal lining and worsen the symptoms of IBD [90] by inhibiting the production of prostaglandins, which are hormones that play a role in pain and inflammation. Other studies have shown that prostaglandins play a protective role in the gut lining, helping to maintain the integrity of the intestinal wall and promoting healing. The inhibition of prostaglandin production by NSAIDs exacerbates inflammation and irritation, can increase the risk of flare-ups, and can worsen symptoms in people with IBD [91]. **(VI)** Other lifestyle changes: Several studies have reported that other factors including adequate sleep (7–9 h of sleep are recommended per night) and limiting alcohol consumption play an important role in maintaining overall health and can also help to reduce stress and inflammation [92,93,94,95]. Thus, there is still a considerable need to improve the treatment of IBD, and significant research efforts are underway to address this unmet need for patients with IBD.

## 3. The Role of Calprotectin in the Inflammatory Response

The S100 protein family, consisting of 25 known members, is the principal participant in inflammation, including S100A8 and S100A9, which are part of the Ca^2+^-binding EF-hand superfamily, characterized by an α-helical structure with two helix–loop–helix Ca(II)-binding EF-hand motifs. Human calprotectin is encoded by two different genes, S100A8 (five exons) and S100A9 (three exons), located on chromosome 1q21.3, which comprise 93 and 113 amino acids, with molecular weights of 10.8 and 13.2 kD, respectively. Calprotectin is a calcium- and zinc-binding heterodimeric molecule that preferentially forms stable heterodimers, which is indispensable for the basic biological functions of cells [96,97]. Calprotectin is a protein that is released by white blood cells and expressed and stored primarily in innate immune cells such as neutrophils [97,98,99], monocytes [100], dendritic cells [101], and activated macrophages [102] in response to inflammation in the gut, making it a useful marker for the diagnosis and monitoring of a wide range of inflammatory conditions including inflammatory arthritis, juvenile rheumatoid arthritis, inflammatory bowel disease [96], transplantation, cystic fibrosis, islet inflammatory response, severe forms of glomerulonephritis, skin stresses, inflammation of the uterine cervix, cardiovascular diseases (CVDs), infections, periodontitis, autoimmune synovitis, peritonitis, microcirculatory defects in diabetic nephropathy, and autoimmune diseases such as juvenile dermatomyositis [103,104]. Therefore, targeting calprotectin is the best choice for ameliorating the inflammation, as several studies have shown, by indirectly or directly targeting the calprotectin protein. For example, the indirect targeting of calprotectin using a variety of specific agents, including antitrafficking therapies, gut-selective biologic agents, selective Janus Kinase (JAK) inhibitors, and anti-interleukin agents, has shown promise in treating IBD [105,106]. Ongoing research in this area is expected to lead to further breakthroughs in the development of these targeted therapies, providing hope for those who have not responded to conventional treatments. Additionally, therapies that directly target the S100A8 and S100A9 proteins have showed promise in disorders associated with inflammation, indicating their superiority to conventional therapies [107]. Reports have shown that, in the presence of Zn^2+^ and Cu^2+^, oral quinoline-3-carboxamide tasquinimod binds to S100A9 and the S100A8/A9 complex and effectively inhibits the interaction between S100A9 and TLR4 or RAGE (one of the cognate receptors for calprotectin) [108]. In S100A9-dependent animal models, this inhibition results in a reduction in TNF release in vivo [109]. These medications have shown promise in the treatment of systemic lupus erythematosus (SLE) [110], type 1 diabetes [111], and multiple sclerosis [112]. A recent study by Euni Cho et al. showed that a recombinant colon-targeted peptide (TWYKIAFQRNRK, designated, rCTS100A8/A9) had a protective effect against intestinal inflammation and tumorigenesis in acute/chronic colitis and against colitis-associated CRC development; it showed promise in reducing damage and inflammation in the guts of mice with a condition similar to IBD-related colon cancer [113].

Additionally, other reports have shown that blocking the secretion or activity of soluble S100A8/A9 complex could be a promising therapeutic approach in surviving patients, who had lower levels of S100A8/A9, compared to non-survivors. The blocking has the potential to prevent liver injury and bacterial dissemination in the early stages of sepsis and endotoxemia [114]; it also alleviates organ injury by reducing tissue damage in the lungs during tuberculosis [115] and lung inflammation and mitigates associated lung diseases during infection with the influenza A virus (IAV) [116] in cases of biofilm-infected persistent wounds. The local targeting of S100A8/A9 could serve as an adjuvant immunotherapy [117] to prevent the destruction of cartilage and bone in RA patients. The administration of an anti-S100A9 antibody has been shown to improve clinical scores by 50% in RA patients [118,119]. Other reports have shown that S100A9 is involved in asthma via initiating and intensifying neutrophilic inflammation, contributing to the disease process [120], and playing a protective role in asthma-related airway hyper-responsiveness by inhibiting the contraction of airway smooth muscles [121]. Other reports have shown that S100A8 and S100A9 are highly expressed in eosinophils during colonic inflammation, contributing to tissue repair. This highlights the potential of eosinophil-mediated pathways as new therapeutic targets for colonic inflammation and repair, particularly in IBD.

The involvement of S100A8, S100A9, and the S100A8/A9 complex in the development of inflammatory diseases suggests that targeting these proteins could be a potential treatment approach, and more evidence is required before their widespread application in clinical practice. Therefore, it is crucial to gain a comprehensive understanding of the specific biological functions, stages, and molecular mechanisms of S100A8 and S100A9 in different inflammatory diseases.

Our team showed in a clinical trial (NCT04749576) that saffron modulates calprotectin levels in patients with UC, suggesting that saffron directly or indirectly affects calprotectin levels or activity through mechanisms that are yet to be investigated. Indeed, we consistently observed a decrease in fecal calprotectin levels in many patients with UC receiving saffron [122].

## 4. Preclinical Observations Regarding the Use of Saffron as a Therapeutic Agent

Saffron is a spice derived from the flower of the *Crocus sativus* plant (dried stigmas of *Crocus sativus* L.) that contains a plethora of bioactive compounds such as carotenoids (crocetin, crocin, α-carotene, lycopene, and zeaxanthin), monoterpene aldehydes (e.g., picrocrocin and safranal), monoterpenoids (e.g., crocusatines), isophorones, and flavonoids [123]; it has been traditionally used in Mediterranean and Middle Eastern cuisine. Based on previous studies and our laboratory findings, saffron has shown promise in various therapeutic applications, exhibiting anti-inflammatory, antioxidant, and immunomodulatory properties [124,125]. Crocin, a water-soluble carotenoid, is the primary pigment in saffron and is responsible for its color and anti-inflammatory properties. The crocetin apocarotenoid dicarboxylic acid analogues crocin, picrocrocin, and safranal are regarded as the most notable bioactive molecules of the saffron [126]. Safranal (2,6,6-trimethyl-1,3-cyclohexadiene-1-carboxaldehyde), a volatile compound that gives saffron its characteristic aroma, has been shown to have antioxidant and anti-inflammatory properties. The detailed pharmacological effects of saffron and its bioactive compounds are presented in Table 1.

Hence, it is important to note that, while these mechanisms have been proposed based on preclinical findings, the precise ways in which saffron exerts its therapeutic effects are still being investigated. Therefore, further research is necessary to gain a comprehensive understanding of the underlying mechanisms and to validate the therapeutic potential of saffron in different contexts.

## 5. Clinical Observations from Using Saffron as a Therapeutic Agent

Inflammation is a key factor in the development and progression of IBD. Several studies have shown that saffron extract can reduce inflammation and oxidative stress in experimental colitis models by suppressing nitric oxide production, iNOS, and COX-2 [147]; modulating immune cells, potentially playing a protective role in IBD [160,161]; reducing inflammation; altering the gut microbiota composition; and preventing the depletion of short-chain fatty acids in mice with experimental colitis, potentially improving colitis symptoms [162]. It also improves antioxidant factors and reduces disease severity in patients with UC [163]. Studies have shown that saffron extract may have a beneficial effect in treating IBD [161,162,163]. Multiple preclinical studies have demonstrated that saffron extract can reduce colonic inflammation and prevent intestinal damage in mice with experimental colitis [122,161]. Additionally, saffron extract has been shown to decrease inflammatory markers in the blood of patients with UC [161,162,163]. A clinical trial involving patients with UC treated with saffron extract for 8 weeks showed promising results [23]. Further clinical trials are needed to fully evaluate the efficacy and safety of saffron in treating IBD; however, these preclinical and clinical findings suggest that saffron may be a promising therapeutic agent for the treatment of IBD. The effects of saffron in various diseases were closely examined in different clinical trials carried out from 2011 to date; most of these were rheumatoid arthritis-associated, as shown in Table 2.

## 6. Saffron Is Safe and Cost-Effective and Has Minimal Potential Side Effects

Saffron and its components (as a natural plant extract) are generally considered safe, beneficial, and tolerable with a wide therapeutic index, when consumed through food or at low doses [191,192]. The lethal dose (LD50) values of saffron suggest that a daily intake of up to 1.5 g is considered safe, while 5 g per kilogram of body weight is considered toxic, and 20 g per kilogram of body weight is considered lethal [193,194]. Overall, in our saffron clinical IBD study, saffron extract at doses of saffron aqueous extract ((50, and 100 mg/kg/day), (Sina Pajoheshan (sinapharmaco.com) provided saffron), exhibited potential protective effects. Various authentic sources (https://www.iherb.com/; https://www.vitacost.com/ and https://nutricost.com/, accessed on 18 June 2024) were used to obtain information about saffron (NFS-02854 or Saffr’Activ^®^) for evaluating and estimating the cost range. Interestingly, the cost of saffron ranged from USD 0.34 to USD 0.11 per 88.5 mg, suggesting that saffron is indeed cost-effective compared to conventional drugs used for IBD—UC/CD; detailed information on the cost of drugs is presented in Appendix A. Studies have also shown that high doses of saffron (5000 mg/kg) cause side effects such as low RBC and WBC counts and hemoglobin levels in BALB/c mice following five weeks of oral ingestion [195]. Other studies have shown its effect on the activity of liver enzymes such as aspartate aminotransferase (AST) and alanine aminotransferase (ALT) at high doses [196]. Other reports have shown that large amounts of saffron (500, 1000, or 2000 mg/kg/day) increase the risk of miscarriage (harming the fetus in the first trimester during organogenesis) and have negative effects on nursing mothers due to potential kidney damage in their neonates [197,198]. Furthermore, high saffron doses have various side effects such as nausea, vomiting, headache, dizziness, diarrhea, dry mouth, changes in appetite, allergic reactions (in rare cases), premenstrual syndrome, postpartum depression, sleep disorders, and snacking behavior [199,200]. Reports have shown that animal models (pregnant BALB/c mice) that received different doses (0.2%, 0.4%, and 0.8%) of aqueous saffron extract in the third trimester of the gestational period showed a significantly decreased placental weight, mean fetal weight, and biparietal diameter, and an increased rate of dead fetuses. Altogether, saffron showed various effects on the development of embryos [201,202]. The teratogenic effects of saffron have been described in detail [203]. Additionally, saffron slightly affects hematological and biochemical parameters but not in a clinically significant manner [192]. Other reports have shown that increased Na+, blood urea nitrogen (BUN), and creatinine levels detected in animals at doses of 200–400 mg of saffron are indicative of kidney dysfunction. Similarly, histopathological changes such as the degeneration of epithelial cells lining the proximal and distal convoluted tubules were observed after saffron treatment [192]. Studies have shown that saffron interacts with certain medications, including antidepressants and blood thinners [195]. Other reports have demonstrated that saffron extract has selective toxic and preventive effects on cancer cells, with low toxicity against non-cancerous cells, and, as such, may be a potential agent for cancer treatment and prevention [204,205].

Furthermore, several studies have demonstrated that saffron and its bioactive compounds such as crocetin, dimethylcrocetin, and safranal interact with DNA at the molecular level and potentially offer antitoxic and anticancer properties [206,207], with antiproliferative, antimigration, and antiadhesion effects on breast cancer [154,208,209], cervical adenocarcinoma [210], prostate cancer [211], glioblastoma, rhabdomyosarcoma cell lines [212], and kidney and bladder cancer cell lines [213].

## 7. Perspective on Saffron Use as an Adjuvant Therapy for Inflammatory Bowel Disease

The perspective of this review article is based on the evidence reported from preclinical and clinical studies showing that saffron and its main pharmacologically active components, including crocin, crocetin, and safranal, may be potential therapeutic agents or adjunct therapeutics due to their anti-inflammatory, antioxidant, and immunomodulatory roles, giving them the potential to mitigate, improve, and manage inflammatory-associated diseases by acting as antihypertensive, antiatherogenic, hypolipidemic, neuroprotective, and anticancer agents [15,127,141,183,214,215,216,217,218,219,220]. Studies have reported that saffron pre-treatment alters the gut microbiota composition, prevents the depletion of short-chain fatty acids (SCFAs) in mice with experimental colitis, improves colitis symptoms by suppressing nitric oxide (NO) and COX-2 production [147], and lowers CRP and TNF levels [221] in mice with dextran sodium sulfate (DSS)-induced colitis [160,162,222].

Several lines of evidence have reported the potential mechanism of saffron and its bio-active compounds against Inflammatory Bowel Disease (IBD) through multiple mechanisms, such as the fact that saffron has anti-inflammatory properties that help reduce cytokine production and inhibit pathways such as NF-κB, which is crucial in the inflammatory response [161,221,223]; its antioxidant activity neutralizes free radicals, protecting intestinal tissues from oxidative damage [163], and modulates the immune system by promoting regulatory T cells while suppressing pro-inflammatory cells [224]. Additionally, saffron influences the gut microbiota composition, enhancing beneficial bacteria and reducing harmful ones [162]. These combined effects contribute to maintaining intestinal barrier integrity, reducing inflammation, and preventing IBD progression.

Inflammation is the body’s first-line defensive response; however, chronic or excessive inflammation can contribute to the development of various inflammatory diseases (autoimmune disorders, cardiovascular diseases, and chronic inflammatory conditions such as rheumatoid arthritis and IBDs such as CD and UC). Studies have reported that calprotectin (S100A8/A9), a calcium-binding protein produced by immune cells (neutrophils and monocytes/macrophages) in response to inflammatory stimuli, serves as an important biomarker of inflammation and is commonly measured in clinical settings to assess the presence and severity of inflammatory conditions. Elevated levels of calprotectin are linked to an inflammatory response and tissue injury in the gastrointestinal tract. Therefore, the tight regulation of calprotectin in ongoing research is the key to fully understanding the complex mechanisms underlying inflammatory stimuli and their regulation as a potential therapeutic target strategy (treatment in various inflammatory conditions) to mitigate inflammation and improve inflammatory diseases’ outcomes. Studies have shown that saffron and its bioactive compounds may have an inhibitory effect that leads to calprotectin levels’ reduction in the colon of rats with induced colitis via the inhibition of the release of several pro-inflammatory cytokines, including IL-6, TNF-alpha [224,225], and calprotectin as we previously reported in patients with IBD and in DSS-colitis mice models [160]. Therefore, how saffron and its bioactive compounds act and structurally interact with calprotectin and how it reduces the level of inflammation is still not clear, and limited research is available. We speculate that saffron directly affects the calprotectin levels or activity by binding its calcium- and zinc-binding sites. Furthermore, the molecular mechanisms underlying saffron’s modulation of calprotectin and the inflammatory response through its bioactive compounds need further research to elucidate the precise mechanisms underlying its therapeutic effects and to explore its clinical translation. The structural interactions and the key signaling pathways (nuclear factor-kappa B (NF-κB)) and mitogen-activated protein kinases (MAPKs) associated with saffron and calprotectin in inflammatory diseases need to be explored. The implications of the interplay between saffron and calprotectin in IBD management are significant, and the potential synergistic effects of saffron and/or its bioactive compounds as an adjuvant therapy in combination with conventional IBD treatments (such as biologics or immunosuppressants) has the potential to improve clinical outcomes, reduce disease activity, and prevent relapse for IBD. In addition, saffron is cost-effective, safe as a natural product, and a promising therapeutic source for multiple ailments, with potential benefits over current medications as an alternative addition for inflammatory diseases with fewer side effects [226,227].

In summary, continued research in these areas (such as the direct inhibition of S100A8/A9 expression, neutralization (in gut), the inhibition of S100A8/A9-induced inflammatory responses, and S100A8/A9-mediated signaling inhibition) hold promise for advancing our understanding of S100A8/A9’s biology and mechanistic insights. Preclinical and clinical studies assessing saffron’s therapeutic potential will improve treatment strategies in a wide range of inflammatory diseases associated with calprotectin, including IBD.

## Figures and Tables

**Figure 1 nutrients-16-02353-f001:**
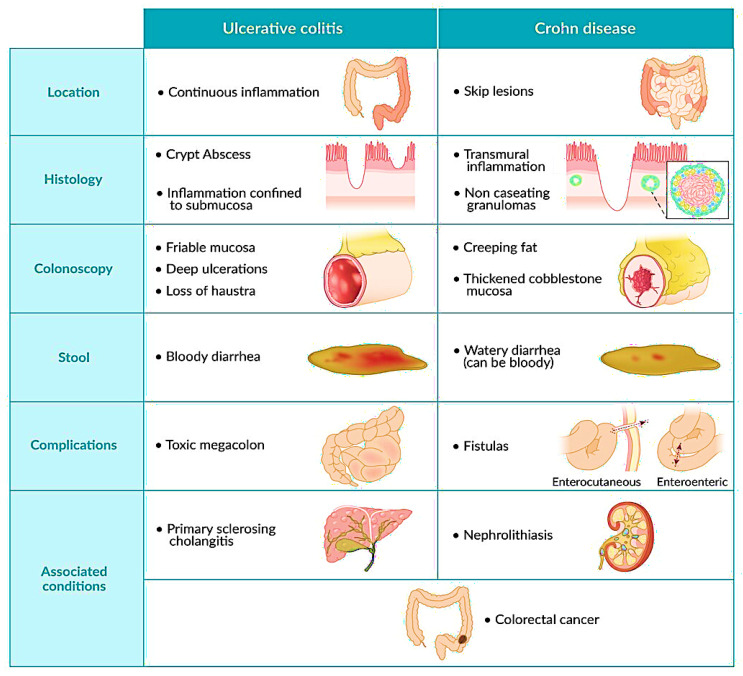
Summary of the clinical features commonly associated with CD and UC: the detailed clinical differences between CD and UC such as the location, histology, colonoscopy, stool complications, and associated conditions such as colorectal cancer. Adopted with permission from the website AMBOSS.com© (https://www.amboss.com/us/knowledge/crohn-disease, accessed on 18 June 2024).

**Table 1 nutrients-16-02353-t001:** Saffron and its bioactive compounds with a variety of pharmacological effects.

Bioactive Compound	Pharmacological Effect	Health Benefits	References
Crocetin	(1) Antioxidant, anti-inflammatory, and immunomodulatory (2) Increases the mRNA expression of anti-inflammatory cytokines (TGF, IL-10, and IL-4) (3) Inhibits COX-2 and NO production in macrophages by modulating the crosstalk between the MEK1/JNK/NF-B/iNOS pathway and the Nrf2/HO-1 pathway (4) Inhibits platelet activity and thrombosis by partially inhibiting calcium elevation in stimulated platelets (5) Stimulates Treg and Th1 cell differentiation by reducing MCP-1 and IL-8 expression and inhibiting immune cell adhesion and infiltration (6) Regulates the Keap1/Nrf2 signaling pathway via inhibition of apoptosis (7) Reduces reactive oxygen species production and cell apoptosis in skin cells (8) Partially agonizes and selectively desensitizes the TRPA1 channel (9) Reports showed that it inhibits *Escherichia coli* cell growth by binding and inhibiting ATP synthase	(1) Cardioprotective/hepatoprotective (2) Reduces diabetic complications (3) Induces neurite growth and facilitates the recovery of motor and sensorimotor functions after spinal cord injury (4) Studies have reported it acts as an enhancer for learning memory and reduces neuropathic pain after spinal nerve transection in rats (5) Shows potential for antiaging, antihypertensive, and anticataract effects (6) Studies have shown neuroprotective effects in ocular diseases, such as age-related macular degeneration, glaucoma, and diabetic maculopathy, and Alzheimer’s and Parkinson’s diseases (7) Protects against UV-A-induced skin damage (8) Exerts analgesic properties (9) Effectively improves insulin sensitivity and related disorders in fructose-fed rats, suggesting its potential as a preventive strategy for insulin resistance-associated diseases	[127,128,129,130,131,132,133,134,135,136,137,138,139,140,141,142,143,144,145,146,147,148,149,150,151,152]
Crocin	(1) Higher solubility and safety, and enhanced inhibitory effects on tumor cell proliferation via inhibiting the angiogenesis-VEGF/VEGFR2 signaling pathway	(1) Potential in treating bone and cartilage diseases (2) Potential for managing diseases associated with abnormal blood vessel growth and inhibiting the growth of human cancer cells in vitro	[150,151,152,153,154,155]
Safranal	(1) Suppresses free radical production and increases antioxidant activity (2) Reduces cardiac injury markers (LDH and CK-MB) caused by sympathetic hyperactivation and decreased TNF-α l, PTGS2, MMP9, and pRELA levels in a model system (3) Potential as a preclinical candidate drug against NLRP3 inflammasome-triggered chronic inflammation by suppressing IL-1 release and preventing ASC oligomerization. (4) Increases the IFN-/IL-4 ratio (5) Reduces oxidative stress and inflammation in a high-fat diet and low-dose streptozotocin-induced type 2 diabetes rat model (6) Reduces inflammation and pain in mice with dextran-sulfate-sodium-induced colitis by suppressing nitric oxide production, iNOS, and COX-2	(1) Reduces diabetic complications (2) Protective effects on ischemic reperfusion injury (3) Cardioprotective (4) Potential therapeutic applications for asthma and allergic reactions (5) Suggests potential therapeutic effects in inflammatory diseases associated with Th1/Th2 imbalance (6) Shows gastro-protective effects against indomethacin-induced gastric ulcers, with similar effects to lansoprazole	[132,133,134,135,144,145,146,147,148,149,156,157,158,159]

Note: CCl4 (carbon tetrachloride); MSCs (mesenchymal stem cells); TGF (transforming growth factor-β); IL-4/10/1 (interleukin-4/10/1); COX-2 (cyclooxygenase-2); NO (nitric oxide); MEK1 OR MAP2K1/JNK/NF-B/iNOS Nrf2/HO-1 (MAP2K1—mitogen-activated protein kinase kinase 1; c-Jun N-terminal kinase; nuclear factor kappa-light-chain-enhancer of activated B cells; inducible nitric oxide synthase; nuclear factor erythroid 2-related factor 2; heme oxygenase-1); LDH (lactate dehydrogenase); CK-MB (creatine kinase-MB); TNF-α (tumor necrosis factor-alpha); NLRP3 (NOD-like receptor family pyrin domain-containing protein 3); ASC (apoptosis-associated speck-like protein containing a CARD (caspase recruitment domain)); OVA (ovalbumin); LTC4 (leukotriene C4); VEGF/VEGFR2 (vascular endothelial growth factor/vascular endothelial growth factor receptor 2); Treg (regulatory T cells); Th1/Th2 (T helper type 1/type 2 cells); IFN (interferon); Keap1 (Keap1 (Kelch-like ECH-associated protein 1); Nrf2 (nuclear factor erythroid 2-related factor 2); TRPA1 (transient receptor potential ankyrin 1); IBD (inflammatory bowel disease); and UC (ulcerative colitis).

**Table 2 nutrients-16-02353-t002:** The detail of saffron clinical trials.

Role of Saffron	Country	Year	Concentration	Participants	Clinical Trial #	Ref.
	**SLEEP QUALITY**					
Reports have shown that saffron intake was associated with improvements in sleep quality in adults with self-reported sleep complaints.	Belgium	2021	15.5 mg per day for 6 weeks	34	NCT04750681	[164]
	Australia	2020	14 mg twice daily for 28 days	33	ACTRN12619000863134	[165]
	**NEUROPSYCHIATRIC CONDITIONS**					
Several lines of evidence have shown that supplementation with saffron showed potential SSRI-like activity and neuroprotective properties, implying that saffron could serve as a safe adjunctive medication to alleviate symptoms, particularly in MDD and postpartum depression, with a notable impact on anxiety disorders and a minimal occurrence of side effects.	Australia	2019	14 mg b.i.d. for 8 weeks	72	NA	[166]
	Australia	2018	14 mg b.i.d. for 8 weeks	40	ACTRN12617000155392	[167]
	Australia	2017	28 mg/day and 22 mg/day for 4 weeks	121	NA	[168]
	Iran	2017	30 mg/day for 6 weeks	30	NA	[169]
	Iran	2017	15 mg twice daily for 6 weeks	34	NA	[170]
	Iran	2016	50 mg twice daily for 12 weeks	54	NA	[167]
	Iran	2015	30 mg/day and 15 mg b.i.d. for 4 weeks	23	IRCT20130418013058N11	[171]
	Iran	2005	30 mg/day (b.i.d.) for 6 weeks	20	NA	[172]
	Iran	2005	30 mg/day capsule for 6 weeks	20	NA	[173]
	Australia	2020	28 mg daily for 6 weeks	31	ACTRN12621000501842	[174,175]
Studies have shown that saffron exhibited efficacy equivalent to methylphenidate in treating ADHD in children, suggesting its potential as a candidate for ADHD therapy due to its ability to impact both monoaminergic and glutamatergic systems, yielding satisfactory outcomes.	Iran	2019	20–30 mg/day for 6 weeks	27	IRCT201701131556N94	[176]
Other reports have shown saffron to be both safe and effective in the short-term for individuals with mild-to-moderate AD, attributed to its ability to inhibit the aggregation and deposition of amyloid β in the human brain, thereby potentially treating neurodegenerative damage caused by oxidative stress.	Iran	2009	30 mg/day (15 mg twice per day)	27	IRCT138711051556N1	[177]
	Iran	2010	30 mg/day for 16 weeks	23	NA	[178]
	**CARDIOVASCULAR EFFECTS**					
Saffron, possessing antioxidant, anti-inflammatory, antihyperlipidemic, hypotensive, and weight-lowering properties, can aid in supporting cardiovascular health and ameliorating symptoms in atherosclerosis patients, including physical disability, sexual dysfunction, and psychological disorders, enhancing quality of life.	Iran	2014	30 mg/day capsule for 6 weeks	22	NA	[179]
	Iran	2022	100 mg/day for 6 weeks	33	NA IRCT201511192017N25	[180]
	**METABOLIC DISORDERS**					
Saffron’s potent antidiabetic, antiobesity, hypotensive, and hypolipidemic effects suggest its potential importance in managing MetS, with studies demonstrating improvements in FBG, hemoglobin A1C, glycemic status, lipid profile, oxidative status, and liver function tests in diabetic profiles.	Iran	2022	100 mg/day for 8 weeks	35	NA	[181]
	Iran	2014	30 mg daily for 12 weeks	44	NA	[182]
	**RHEUMATOID ARTHRITIS**					
A study showed the potential benefits of saffron supplementation in enhancing disease activity and clinical outcomes in RA by decreasing inflammatory ILs, highlighting its anti-inflammatory properties and ability to alleviate acute and chronic pain.	Iran	2020	100 mg/day for 12 weeks	31	NA	[183]
	**REPRODUCTIVE HEALTH**					
Studies have explored saffron’s aphrodisiac effects in men, indicating improvements in erectile function and overall sexual health, especially in diabetes. Similarly, in women, saffron has proven effective in alleviating sexual dysfunction and relieving symptoms of premenstrual syndrome, dysmenorrhea, and irregular menstruation, potentially modulating the secretion of steroid hormones.	Iran	2022	15 mg twice daily for 2 weeks	34	IRCT20090117001556N110	[184]
	Iran	2015	1% topical saffron gel	25	IRCT ID: 201404071769N1	[185]
	Iran	2008	30 mg/day for 2 menstrual cycles	25	NA	[186]
	**Inflammatory Bowel Disease**					
Reports have shown that saffron supplementation among patients with UC may be effective in improving antioxidant status and reducing disease severity. Our multiple-center IBD clinical trial has suggested that saffron treatment led to a decrease in pro-inflammatory (TNFα, INF-γ, IL-6, IL-2, and IL-17a) and an increase in anti-inflammatory (IL-10 and TGF-β) cytokines, along with reduced fecal calprotectin (CP) and serum CRP levels in patients with mild-to-moderate UC	Iran USA	2020 Presently active	100 mg/daily 50mg/b.i.d	40 62	NA NCT04749576	[163] [122,187]
	**IMMUNOREGULATORY**					
Saffron may have effects on the immune system and hematological parameters.	Iran	2011	100 mg daily for 6 weeks	45	NA	[188]
Saffron may have mental and physical effects in healthy recreationally active adults	Australia	2020	28 mg daily for 6 weeks	31	ACTRN12621000501842	[174]
	**OCULAR DISEASES**					
Saffron supplementation modestly improved visual function in participants with AMD, including those using AREDS supplements. Additionally, saffron supplementation shows promise in slowing down the progression of central retinal dysfunction in ABCA4-related STG/FF.	Italy	2019	20 mg over 180 days	14	NCT01278277	[189]
	Australia and New Zealand	2019	20 mg/day for 3 months	50	ACTRN 12612000729820	[190]

Note: SSRI (selective serotonin reuptake inhibitor), major depressive disorder (MDD), ADHD (attention deficit hyperactivity disorder), Alzheimer’s disease (AD), metabolic syndrome (MetS), FBG (fasting blood glucose), hemoglobin A1C (Hb A1C), b.i.d (twice a day), rheumatoid arthritis (RA), interleukins (ILs), AMD (age-related macular degeneration), including those using AREDS (Age-Related Eye Disease Study), ABCA4-related STG/FF (Stargardt disease and fundus flavimaculatus), and IBD (inflammatory bowel disease), NA (not available, due to multiple reasons—outside USA and other).

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
