# Peer review of "Saffron as a Promising Therapy for Inflammatory Bowel Disease"

_nutrients, 2024, doi:10.3390/nu16142353_

Round 1

Reviewer 1 Report (Previous Reviewer 2)

Comments and Suggestions for Authors

Authors have aptly replied to all comments.

Author Response

Dear Reviewer,

Thank you for considering our manuscript for publication with no comment.

Sincerely,

Hassan Ashktorab, Ph.D.

Professor, Department of Medicine, Cancer center

Adjunct graduate professor of Human Genetic

University of Howard

Reviewer 2 Report (New Reviewer)

Comments and Suggestions for Authors

Dear Author,

I have carefully reviewed your manuscript and would like to provide some feedback for improvement:

1. In the "2. Current treatments and their limitations" section, the ordering of the treatments for inflammatory bowel disease (IBD) is inconsistent. The ASAs are numbered, while the subsequent treatments use letters. I would recommend standardizing the formatting and using numbered items throughout for consistency.

2. In the same section, line 85 lists a range of drugs used for IBD treatment, where E (Surgery) and F (Lifestyle changes) are not pharmacological treatments. It is suggested to revise this list to accurately reflect non-pharmacological interventions.

3. In Table 1, there is a minor error in the References section where "151-153]" should be corrected to "[151-153]".

4. In the Note on page 10, consider revising "ulcerative colitis (UC)" to "UC (ulcerative colitis)" for better clarity and consistency.

5. The reference citations in the section from line 363 to 367 Studies have shown that saffron extract may have a beneficial effect in treating IBD [1]. Multiple preclinical studies have demonstrated that saffron extract can reduce colonic inflammation and prevent intestinal damage in mice with experimental colitis [1]. Additionally, saffron extract has been shown to decrease inflammatory markers in the blood of patients with UC [1].”appear to be incorrect. Please ensure the accuracy of the references cited in this part of the text.

6. The paragraph numbering in line 375 should be corrected to "6," and similarly, in line 427, the paragraph numbering should be adjusted to "7" for consistency and clarity.

These revisions will enhance the accuracy and clarity of your manuscript. Thank you for considering these suggestions for the improvement of your work.

Best regards,

Comments on the Quality of English Language

Minor editing of English language required,Revise the grammar and individual expressions carefully

Author Response

  1.  
  2.  
  3. 1. In the "2. Current treatments and their limitations" section, the ordering of the treatments for inflammatory bowel disease (IBD) is inconsistent. The ASAs are numbered, while the subsequent treatments use letters. I would recommend standardizing the formatting and using numbered items throughout for consistency.

 Response: We want to thank to the reviewer for the valuable comment, we added the correct order highlighted in yellow in the  manuscript (Line#95)

  1. In the same section, line 85 lists a range of drugs used for IBD treatment, where E (Surgery) and F (Lifestyle changes) are not pharmacological treatments. It is suggested to revise this list to accurately reflect non-pharmacological interventions.

 Response: Thank you for the suggestion, we revised the manuscript highlighted in yellow (Line#94 and 203)

  1. In Table 1, there is a minor error in the references section where "151-153]" should be corrected to "[151-153]".

  Response: Thank you for the comment, we have revised the manuscript with highlighted in yellow (Table 1).

  1. In the Note on page 10, consider revising "ulcerative colitis (UC)" to "UC (ulcerative colitis)" for better clarity and consistency.

   Response: Thank you for the comment, we have changed  with highlighted in yellow in the manuscript (Table 1)

  1. The reference citations in the section from line 363 to 367 “Studies have shown that saffron extract may have a beneficial effect in treating IBD [1]. Multiple preclinical studies have demonstrated that saffron extract can reduce colonic inflammation and prevent intestinal damage in mice with experimental colitis [1]. Additionally, saffron extract has been shown to decrease inflammatory markers in the blood of patients with UC [1].”appear to be incorrect. Please ensure the accuracy of the references cited in this part of the text.

 Response: Thank you for the comment, we have changed and added the correct references with highlighted in yellow in the manuscript (Line#360-364)

  1. The paragraph numbering in line 375 should be corrected to "6," and similarly, in line 427, the paragraph numbering should be adjusted to "7" for consistency and clarity.

 Response: Thank you for the comment, we have changed the order, highlighted in yellow in the manuscript(Line#372 and 418).

Minor editing of English language required, Revise the grammar and individual expressions carefully

Response: We have used the MDPI English language services (english-81972) and hope they have done perfect job and not need to further revise the grammar in the manuscript.

Sincerely,

Hassan Ashktorab, Ph.D.

Professor, Department of Medicine, Cancer center

Adjunct graduate professor of Human Genetic

University of Howard

Reviewer 3 Report (New Reviewer)

Comments and Suggestions for Authors

Dear Authors,

 the paper presented for review is correct, it contains a few editorial errors (literature - capital letters in titles, hyperlinks not removed) - easy to remove. My comment, however, relates to the novelty of this work, given the numerous review papers on saffron and its use in IBD. I leave it to the Editors to decide on this topic.

Author Response

July 13, 2024

Dear Nutrients Reviewer,

I am pleased to submit our revised manuscript entitled “Saffron as a Promising Therapy for Inflammatory Bowel Disease” for consideration by the Journal of Diabetes Research. We have diligently addressed the reviewers’ concerns and provided comprehensive point-by-point responses.

Thank you for your attention to our work, and we look forward to its publication in your esteemed journal.

Sincerely,

Hassan Ashktorab, Ph.D.

Professor, Department of Medicine, Cancer center

Adjunct graduate professor of Human Genetic

University of Howard

Reviewer 4 Report (New Reviewer)

Comments and Suggestions for Authors

This manuscript summarized the use of saffron and its therapeutic potential towards various inflammatory diseases. The work was not well conceived. The full contents do not fit the title of present manuscript. According to the title of this manuscript, the contents should focus on the underlying beneficial effect of saffron on IBD, and its potential mechanisms of action. And in fact, this whole manuscript described more on the limitations and use of commonly used drugs for treatment of IBD, and less contents on the saffron and its potential use on IBD. It is recommended that authors carefully revised the title of the article or rewrite the content of the whole manuscript to make the content fits the title of the article well.  In addition, there are some formatting issues need to be addressed by the author.

(1) Line 348, The table format of Table 1 is not consistent with that of Table 2, please standardize the format of the table and use a three-line table format for both.

(2) Line 375, The left alignment of the notes section of Table 2 is much better.

(3) Line 475, there is no “Funding Information”, please add funding information.

(4) Line 86, the order of the headings is a bit strange, according to the order of the headings of other contents, the heading of line 85 should be “A”, not “1”.

(5) Line 336-347 has unnecessary spacing, please remove it.

(6) Line 476, please remove the unnecessary spacing.

(7) Page 12, The format of the “IBD” section in Table 2 is not consistent with other contents in Table 2.

(8) Some refs in Table 2 are underlined, please standardize the format.

(9) Table 1 summarized the pharmacological effect of Saffron's bioactive compounds, and the relative health benefit, according to the title of the manuscript, the contents in Table 1 cannot reflect the beneficial potential of Saffron towards IBD. It's better to summarized the literatures related to the IBD and saffron, and include its potential mechanism of action.

(10) Line 10, “Abstracts” section should include a short summary about the potential mechanisms of actions by which Saffron exerts its protective effects against IBD.

Author Response

July 13, 2024

Dear Nutrients Editor,

I am pleased to submit our revised manuscript entitled “Saffron as a Promising Therapy for Inflammatory Bowel Disease” for consideration by the Journal of Diabetes Research. We have diligently addressed the reviewers’ concerns and provided comprehensive point-by-point responses.

Thank you for your attention to our work, and we look forward to its publication in your esteemed journal.

Reviewer 3:

Comments and Suggestions for Authors

This manuscript summarized the use of saffron and its therapeutic potential towards various inflammatory diseases. The work was not well conceived. The full contents do not fit the title of present manuscript. According to the title of this manuscript, the contents should focus on the underlying beneficial effect of saffron on IBD, and its potential mechanisms of action. And in fact, this whole manuscript described more on the limitations and use of commonly used drugs for treatment of IBD, and less contents on the saffron and its potential use on IBD. It is recommended that authors carefully revised the title of the article or rewrite the content of the whole manuscript to make the content fits the title of the article well.  In addition, there are some formatting issues need to be addressed by the author.

  • Line 348, The table format of Table 1 is not consistent with that of Table 2, please standardize the format of the table and use a three-line table format for both.

Response: Thank you for the comment, we have changed the format and have almost same with table 1 and 2, highlighted in yellow in the manuscript (table1,line #345 and table 2,line 371).

  • Line 375, The left alignment of the notes section of Table 2 is much better.

Response: Thank you for the comment, we have changed the format in table 2, highlighted in yellow in the manuscript (table 2,line 371.. Role of Saffron content).

  • Line 475, there is no “Funding Information”, please add funding information.

Response: Thank you for the comment. We added the following

Funding: This project was supported partly by the NCI R01CA258519 (HA) and    National Institute on Minority Health and Health Disparities of the National Institutes of Health under Award Number G12MD007597. The funders had no role in study design, data collection and analysis, decision to publish, or preparation of the manuscript.

  • Line 86, the order of the headings is a bit strange, according to the order of the headings of other contents, the heading of line 85 should be “A”, not “1”.

Response: We want to thank to the reviewer for the valuable comment, yes we agree and added the correct order highlighted in yellow in the  manuscript (Line#95)

  • Line 336-347 has unnecessary spacing, please remove it.

Response: We want to thank to the reviewer for the valuable comment, we have removed the  unnecessary spacing in the  manuscript.

  • Line 476, please remove the unnecessary spacing.

Response: We want to thank to the reviewer comment, we have removed the  unnecessary spacing in the manuscript

(7) Page 12, The format of the “IBD” section in Table 2 is not consistent with other contents in Table 2.

Response: Thank you for the comment, we have changed the format in table 2, highlighted in yellow in the manuscript (Table 2).

(8) Some refs in Table 2 are underlined, please standardize.

Response: Thank you for the comment, we have removed the underline and the revised the manuscript highlighted in yellow (Table 2).

(9) Table 1 summarized the pharmacological effect of Saffron's bioactive compounds, and the relative health benefit, according to the title of the manuscript, the contents in Table 1 cannot reflect the beneficial potential of Saffron towards IBD. It's better to summarize the literatures related to the IBD and saffron and include its potential mechanism of action.

Response: Thank you for the comment, In Table 1, We have mentioned detail about overall pharmacological effects of saffron for different Health Benefits, however we have added” potential mechanisms of actions by which Saffron exerts its protective effects against IBD in the manuscript (line# 430-439).

(10) Line 10, “Abstracts” section should include a short summary about the potential mechanisms of actions by which Saffron exerts its protective effects against IBD.

Response: Thank you for the comment,  we have mentioned detail about potential mechanisms of actions by Saffron against IBD in abstract (line #22-28) and manuscript (line# 430-439).

Please let me know if you need additional information and thank you for your consideration.

Sincerely,

Hassan Ashktorab, Ph.D.

Professor, Department of Medicine, Cancer center

Adjunct graduate professor of Human Genetic

University of Howard

This manuscript is a resubmission of an earlier submission. The following is a list of the peer review reports and author responses from that submission.

Round 1

Reviewer 1 Report

Comments and Suggestions for Authors

The review by Rashid et al. on the potential use of saffron in the therapeutic regimen for IBD is interesting. There are, however, some major points that need to be addressed. Here are some suggestions.

1.       The authors introduce Calprotectin on page 6, without describing its structure, that this protein is a heterodimer of S100A8 and S100A9 EF-hand calcium-binding proteins, and go on with discussing the subunits, without connecting them to calprotectin. Please describe in a concise way the protein at the beginning of the paragraph. There is this very recent work describing the colon-targeting of S100A8/A9 in the context of IBD in animal models: https://doi.org/10.1038/s41401-023-01188-2.

2.       The title is too long and not representative of the review. It may be changed to “ Saffron's potential in Targeting Calprotectin (S100A8/A9) for Mod ulating Inflammation and Immune Response: Insights from Preclinical and  Clinical Evidence

3.       Table 1 is not readable, it should be reformatted into columns and describe only the essentials. Do not put whole texts in the tables. Columns titles can be “disease”, “models”, “bioactive compound”, “dose”, “frequency of administration”, “results”, “references”. Tables should list only important information!A legend should accompany each table.

4.       Table 2: what is the source of information regarding the clinical trials? Clinicaltrials.gov? Please list the study numbers (for example, NCT…).

5.       There are repetitions. For instance, the optimum, toxic and lethal dose of saffron are described twice (end of section 4, and beginning of section 5, on page 12), Please compact…

6.       In the section “5. Saffron is safe, cost-effective and does not have potential side effects at optimum 344 doses:”, no mention of cost-effectiveness is made. The title should be short and show impact, and be representative of the context of the paragraph.

7.       Same for the section “Saffron interaction with other drug:”. This section mixes the interaction of saffron with other drugs and the beneficial effects of saffron as anti-cancer agent…Please put the anti-cancer therapy part in a separate paragraph and dwell more on the anti-cancer role, with appropriate description of studies done and their references. This is an important aspect, which should be described in more detail.

8.       Section 6. Again there is repetition on previously described pathologies. Cardiovascular…., page 13. This paragraph should concentrate on the beneficial role of saffron in IBD and on colon inflammation. Description about the potential use of saffron as part of the therapeutic regimen should be included…any potential side effects? Any interaction with currently used drugs? It should be clear that saffron only cannot cure IBD…but it may help in modulating inflammation, if used at optimal dose (what is the optimal dose, when considering IBD?)

Comments on the Quality of English Language

1.       English should be thoroughly revised by native English person. For instance, “Pre-clinical observations using Saffron as therapeutic agent” should read “Pre-clinical observations regarding the use of Saffron as therapeutic agent

2.       Punctuations and spacing facilitate reading, and therefore, should be used appropriately. Please check text.

3.        Page 7, line 30, why is “However, …” used? It may be substituted with “Importantly,…”

Author Response

May 11, 2024

Dear Editor,

We would like to express our gratitude to the reviewers and the editor for their positive comments on our manuscript. Below, we have provided point-by-point responses to the reviewers' concerns.

Reviewers 1 comments:

The review by Rashid et al. on the potential use of saffron in the therapeutic regimen for IBD is interesting. There are, however, some major points that need to be addressed. Here are some suggestions.

  1. The authors introduce Calprotectin on page 6, without describing its structure, that this protein is a heterodimer of S100A8 and S100A9 EF-hand calcium-binding proteins, and go on with discussing the subunits, without connecting them to calprotectin. Please describe in a concise way the protein at the beginning of the paragraph. There is this very recent work describing the colon-targeting of S100A8/A9 in the context of IBD in animal models: https://doi.org/10.1038/s41401-023-01188-2.

Response: We want to thank the reviewer for the valuable comments, which we addressed in lines 241-252 in the revised manuscript. Additionally, a reference and other details have been added with in lines 274-278 in the manuscript.

  1. The title is too long and not representative of the review. It may be changed to “ Saffron's potential in Targeting Calprotectin (S100A8/A9) for Modulating Inflammation and Immune Response: Insights from Preclinical and Clinical Evidence.

Response: Thank you for the suggestion., We have modified title “Saffron as a Promising Therapeutic Avenue for Inflammatory Bowel Disease and Potential Impact on Calprotectin Levels.

  1. Table 1 is not readable, it should be reformatted into columns and describe only the essentials. Do not put whole texts in the tables. Columns titles can be “disease”, “models”, “bioactive compound”, “dose”, “frequency of administration”, “results”, “references”. Tables should list only important information! A legend should accompany each table.

Response: We have revised Table 1 as requested.

  1. Table 2: what is the source of information regarding the clinical trials? Clinicaltrials.gov? Please list the study numbers (for example, NCT…).

Response: We have revised and added all the available clinical trials numbers in Table 2 as requested.

  1. There are repetitions. For instance, the optimum, toxic and lethal dose of saffron are described twice (end of section 4, and beginning of section 5, on page 12), Please compact…

Response: We have removed repetition and revised Section 5 in the manuscript accordingly.

  1. In the section “ Saffron is safe, cost-effective and does not have potential side effects at optimum 344 doses:”, no mention of cost-effectiveness is made. The title should be short and show impact, and be representative of the context of the paragraph.

Response: We have revised section 5 and added cost-effective sentences as well (that was in section 6). We also revised the title to read as’ “Saffron as a Promising Therapeutic Avenue for Inflammatory Bowel Disease and potential Impact on Calprotectin levels” .

  1. Same for the section “Saffron interaction with other drug:”. This section mixes the interaction of saffron with other drugs and the beneficial effects of saffron as anti-cancer agent…Please put the anti-cancer therapy part in a separate paragraph and dwell more on the anti-cancer role, with appropriate description of studies done and their references. This is an important aspect, which should be described in more detail.

Response: Thank you for your valuable suggestion. Although the topic is not related to context, we have briefly added the anti-cancer role of saffron in Section 5 in the revised manuscript .

  1. Section 6. Again, there is repetition on previously described pathologies. Cardiovascular…., page 13. This paragraph should concentrate on the beneficial role of saffron in IBD and on colon inflammation. Description about the potential use of saffron as part of the therapeutic regimen should be included…any potential side effects? Any interaction with currently used drugs? It should be clear that saffron only cannot cure IBD…but it may help in modulating inflammation, if used at optimal dose (what is the optimal dose, when considering IBD?)

Response: We have revised section 6 and deleted some unrelated information in the manuscript precisely as highlighted in yellow color.

Comments on the Quality of English Language

  1. English should be thoroughly revised by native English person. For instance, “Pre-clinical observations using Saffron as therapeutic agent” should read “Pre-clinical observations regarding the use of Saffron as therapeutic agent
  2. Punctuations and spacing facilitate reading, and therefore, should be used appropriately. Please check text.
  3. Page 7, line 30, why is “However, …” used? It may be substituted with “importantly.”

Response: We have revised the manuscript accordingly.

Sincerely,

Ashktorab
Professor

Reviewer 2 Report

Comments and Suggestions for Authors

The manuscript reviews the clinical efficiency of using saffron in IBD. The review is detailed but does need to address a few comments:

1. The title is very long and not to the point. It should be shortened with focused emphasis on saffron and its clinical importance in IBD.

2. The authors should list clinical studies (if any) with the number of patients recruited for the studies on all available treatments, especially conducted for IBD and their effective dosage values. Also, if the drugs/treatments cause toxicity, what are the dosage limits.

3.  Modify Table 1 to look more concise with important details only instead of complete sentences.

4.  Since the authors have referenced studies more towards the efficacy of bioactive compound of saffron, crocetin, crocin, and others; are authors showing clinical efficiency of crocetin and others as a bioactive compounds or recommending saffron as a medium to obtain these bioactive compounds through foods?

5. The authors list saffron as cost-effective. Can they list a couple of references showing the cost-effectiveness of using saffron or its bioactive compounds?

Comments on the Quality of English Language

The English language needs minor editing. The sentences should be more clear and concise.

Author Response

May 11, 2024

Dear Editor,

We would like to express our gratitude to the reviewers and the editor for their positive comments on our manuscript. Below, we have provided point-by-point responses to the reviewers' concerns.

Reviewer 2 comments:

Comments and Suggestions for Authors

The manuscript reviews the clinical efficiency of using saffron in IBD. The review is detailed but does need to address a few comments:

  1. The title is very long and not to the point. It should be shortened with a focused emphasis on saffron and its clinical importance in IBD.

Response: Thank you for suggesting for changing the title : we  modiied the title as “Saffron as a Promising Therapeutic Avenue for Inflammatory Bowel Disease and potential Impact on Calprotectin levels” highlighted in yellow color.

  1. The authors should list clinical studies (if any) with the number of patients recruited for the studies on all available treatments, especially conducted for IBD and their effective dosage values. Also, if the drugs/treatments cause toxicity, what are the dosage limits.

Response: Thank you, for the valuable suggestion and we have revised and addressed the comment in Table 2.

  1. Modify Table 1 to look more concise, with important details only instead of complete sentences.

 Response: Thank you, for the valuable suggestion and we have revised Table 1 as requested.

  1. Since the authors have referenced studies more towards the efficacy of bioactive compounds of saffron, crocetin, crocin, and others; are authors showing clinical efficiency of crocetin and others as a bioactive compounds or recommending saffron as a medium to obtain these bioactive compounds through foods?

Response: Thank  you for your comments. We have addressed the use of unmodified saffron for IBD in our previous studies, observing its positive outcomes in patients by reducing calprotectin levels. While our focus has been on whole saffron extract, it's worth noting that each individual component possesses its own anti-inflammatory effects, although we have not emphasized this aspect as much.

  1. The authors list saffron as cost-effective. Can they list a couple of references showing the cost-effectiveness of using saffron or its bioactive compounds?

Response: Thank you, for the valuable suggestion and we have revised and added in section 5, highlighted in yellow color.

Comments on the Quality of English Language

The English language needs minor editing. The sentences should be clearer and more concise.

Response: The manuscript has been fully reviewed and edited linguistically.

Sincerely,

Ashktorab
Professor
